# Atopic Dermatitis: Clinical Aspects and Unmet Needs

**DOI:** 10.3390/biomedicines10112927

**Published:** 2022-11-14

**Authors:** Fabio Lobefaro, Giulio Gualdi, Sergio Di Nuzzo, Paolo Amerio

**Affiliations:** 1Dermatology Department of Medicine and Aging Science, University of Chieti Pescara, 66100 Chieti, Italy; 2Department of Medicine and Surgery, University of Parma, 43121 Parma, Italy

**Keywords:** atopic dermatitis, adult atopic dermatitis, clinical features, diagnosis, moderate-severe atopic dermatitis, management, treatment, unmet needs

## Abstract

Atopic dermatitis is a common chronic-relapsing, inflammatory and itchy eczematous skin disorder which occurs in both children and adults. AD pathogenesis is complex and several factors are implicated. Pruritus plays a pivotal role in disease’s burden, significantly worsening atopic patient quality of life by limiting productivity and daily activities. AD diagnosis relies still on the experience of the healthcare professional and there are several unmet needs as for the diagnostic criteria, the management and the recognition of the burden of the disease. In this paper we present an indeep focus on the main clinical features of AD and the major unmet needs that should be addressed in the next research.

## 1. Introduction

Atopic dermatitis is a chronic-relapsing, inflammatory and itchy eczematous skin disorder which affects 15–30% and 2–10% of children and adults, respectively [1].

Disease onset occurs mainly during infancy, and it can persist during adulthood, although adult onset atopic dermatitis is described in about 20–25% of cases [2,3].

AD pathogenesis is complex and multifactorial. Genetic factors are involved with stratum corneum structural protein codifying genes deficiency such as filaggrin. Upregulation of genes which encode for Th2 cytokines, such as IL-4 and IL-13 and polymorphisms of gene which encode for their receptors, is also involved. Th2 axis overexpression determines skin barrier dysfunction with downregulation of key proteins for stratum corneum stability such as filaggrin, locrin, involucrin, corneodesmosin. Decreased production of antimicrobial peptides LL-37 and β-defensin results in an increased risk of *S. aureus* bacterial superinfection [4,5,6]. Additionally, several environmental factors such as diesel exhaust, humidity and skin irritants are implicated.

Pruritus plays a pivotal role in disease’s burden, significantly worsening atopic patient quality of life by limiting productivity and daily activities. The role of the histamine-independent signaling pathway in the itching of patients with AD emerged. 

Skin barrier disruption leads to inflammatory cytokines release, including CXCL1 which recruits neutrophils. Neutrophils trigger the release of CXCL10 and CXCR3-expressing sensory neurons that transmit itch signals [7].

Chronic itch is characterized by infiltration of TSLP receptor-expressing basophils, which release IL-4 and CD4+ T cells, which release IL-4, IL-13 and IL-31.

IL-4, IL-13 and IL-31 are implicated in the genesis of pruritus, particularly by their involvement stimulating specific sensitive neurons through a JAK-dependent mechanism; also, IL-31 stimulates sensitive nerve fibers elongation. IL-31 binding to its receptors expressed on sensory nerve fibers induce pruritus via TRPV1 and/or TRPA1 channels. 

Furthermore, IL-31 binds to IL-31RA, a combination which is expressed on keratinocytes and stimulates LTB4 secretion. LTB4 binds to BTL1 receptors, which are localized on sensory nerve fibers.

Sensory neurons are subsequently activated and drive itch sensations and scratching behaviors [4,6,8,9].

A more recent study [10] added new findings in AD pruritus knowledge. Authors discovered that AD inflammation triggers an early phenotypic switch in patients circulating basophils, selecting a specific subset of basophils that display enhanced expression of FcεRI and CD203c.

This subset of IgE-R+ basophils release LTC4 via antigen-specific IgE stimulation. LTC4 binds its receptor CysLTR2, which is expressed on sensory nerve fibers, activating sensory neurons which transmit itch signals, driving scratching behaviors and pruritus during flare-ups [11].

Clinical features include eczematous lesions localized by age, crusts and excoriations, lichenification, cutaneous xerosis, and intense itch up to erythroderma in severe cases. Furthermore, a lot of psychosocial and systemic comorbidities are also associated with AD.

Diagnosis is clinical, and possibly histopathological in doubtful cases. The application of diagnostic criteria is currently limited to inclusion in RCTs or epidemiological studies.

Management depends on the severity of the disease: in mild cases the use of topical medicaments is adequate, while in moderate–severe cases, systemic drugs are needed.

Despite recent advances in knowledge of AD pathogenesis and treatment, there are several unmet clinical needs in patients with atopic eczema.

## 2. Clinical Features

It is known that AD is the primum movens for atopic march developing, with progression in clinical manifestations of AD, asthma, rhinoconjunctivitis and food allergies.

Children with atopic dermatitis have an increased risk of developing allergic asthma [12]. Moreover, AD children with severe sensitization and severe diseases have an increased risk of developing allergic asthma [13]. However, a systematic review of 13 cohort studies showed that only 1 in 3 children with AD develop asthma, and therefore the risk is lower than assumed [14]. More recently, risk factors for the development of allergic asthma and respiratory disease in patients with AD background have been identified: they are represented by greater disease severity, early onset [15] and elevated specific IgE with early sensitization [16], despite the few studies conducted on clinical predictive value of IgE.

Atopic dermatitis is characterized morphologically by erythematous intensely itchy papules or patch with excoriations and exudation. The distribution of lesions varies according to patient’s age and disease activity. Chronic lesions are characterized by lichenification.

Moderate–severe AD is a condition characterized by a multidimensional impact on sleep, psychoemotional disorders, limitations of social life and work impairment, with consequent reduction in quality of life of patients and their families.

Itch is the most typical and common symptom associated with AD, and is often so bothersome that it interferes with normal daily life activities such as work, study and sleep [1,17]. It usually worsens at night and is aggravated by allergens, reduced humidity, sweating, skin irritants. It causes scratching, excoriations and lichenification. It correlates well with disease severity.

Cutaneous pain has also recently been identified among the top 3 most frequent symptoms in patients with AD, including itching and sleep disturbances: it contributes to aggravating disease burden [18,19].

A high prevalence of sleep disturbance, to which the itch–scratch cycle largely contributes, has been reported; furthermore, AD is associated with emotional stress and psychosocial discomfort [20].

AD phenotypes are typically classified as it follows.

### 2.1. Infantile Phase

The infantile phase starts in the first months with eczematous, lesions predominantly involving the cheeks and extensor surfaces of trunk and limbs, sparing the central face and diaper area. Lesions consist of itchy eczematous papules or patch and may become exudative and crusted as result of rubbing. Itch is intense with nocturnal exacerbation and may cause sleep disturbance. Infants may appear irritable, whimper and perform movements equivalent to scratching, such as rubbing and twisting the body. Disease severity mitigates towards the age of 2.

### 2.2. Childhood Phase

Childhood phase ranges from 2 to 12 years. Acute lesions mitigate, and eyelid and neck involvement occurs. Flexural area involvement of both upper and lower limbs is typical. Cutaneous xerosis and signs like Dennie–Morgan folds are more prominent. Itching symptoms are intense and can impair school performance. During this phase, the disease may have a significant psychological impact on the child.

### 2.3. Adult and Adolescent Phase

Adult and adolescent phases range from 12 years. Involvement of face, neck and upper trunk is common, much as periorificial and acral involvement is often observed. Hands may be affected by chronic eczema: diagnosis could represent a challenge in these cases [21]. Although it often afflicts patient since infancy, adult-onset AD is described in 20% of cases.

Xerosis and lichenification are predominant due to the long duration of disease. Adults with erythroderma or poorly responsive to drugs should undergo skin biopsy to exclude cutaneous T-cell lymphoma [22].

Lesion distribution differs by age onset, particularly in pre-adult-onset AD flexural involvement is more common, while trunk is typically involved in adult-onset AD. Furthermore, flexural sites are more commonly affected at first in pre-adult-onset AD while head and neck in adult-onset AD [23].

In a 59-case series, several atypical morphological variants of AD were identified; most common were nummular lesions, followed by prurigo-like, seborrheic dermatitis-like and follicular lesions [24].

Several clinical variants of AD could be distinguished in adult patients as follows [25].

### 2.4. Head and Neck Dermatitis

This is the most typical form of AD in adults and is characterized by head-and-neck involvement, including lips and eyelids, in association or not with lesions classically distributed in flexural areas.

Portrait dermatitis is a specific form of head and neck AD in which lesions affect seborrheic areas of the trunk with a distribution resembling a bust sculpture. This latter is observed more frequently in adolescents, and is believed to be related to Malessenzia yeast sensitization.

### 2.5. Hand Eczema

It is estimated that about 60% of AD patients are affected by chronic hands eczema: this latter should rise the suspicion of AD in the clinician.

Three involvement pattern could be distinguished:Acute relapsing dyshidrotic eczema, characterized by recurrent relapsing vesicle-bullous itchy rash with lesions embedded in the epidermis on an eczematous exudant background. It involves mostly palms and soles. Some authors consider it a nosological entity apart from AD.Chronic hyperkeratonic eczema, often indistinguishable from irritative contact dermatitis, it mainly affects the dorsal surface of the hands.Chronic dry pulpitis, in which erythematous-desquamative involvement affects only the distal phalanges.

### 2.6. Lichenoid AD

This variant occurs mostly in melanodermic individuals of African or Latino-American ethnicity, with itchy flat non-purplish erythematous papules involving both upper and lower limbs. Skin biopsy is needed to rule out Lichen Planus.

### 2.7. Nummular Eczema

This variant is historically linked to Asian AD phenotype.

Nummular type AD is characterized by eczematous coin-like shaped exudant excoriated patches, involving mostly lower limbs [26]. Although associated with AD, nummular eczema could have further etiologies, which may be infectious or allergic: this latter should be excluded by patch testing.

### 2.8. Follicular AD

It is observed mostly in Asian AD patients with intrinsic phenotype [26].

It occurs with follicular prominence on the trunk, back, abdomen and proximal extremities, with intense itch [22].

### 2.9. Atopic Prurigo

Prurigo-like AD affects about 30% AD adult patient and occurs with papulo-nodular intensely itchy excoriated lesions, localized mostly on lower limbs [27].

In these cases, a cutaneous biopsy should be used to rule out other severe conditions such as nodular subtype of bullous pemphigoid or cutaneous T-cell lymphoma, especially in elderly patients.

### 2.10. Psoriasiform AD

Psoriasiform AD occurs with widespread lesions localized on trunk and limbs, with flexural involvement. It is more frequently observed in Asian populations. Diagnosis can be challenging, and sometimes an overlap between AD and psoriasis can be observed.

In this latter, the presence of psoriatic lesions involving the scalp, knees or nails, in association with eczematous AD site-specific lesions, is typical [28]. It is often difficult to treat and poorly responsive to conventional therapies. Th17 and Th23 axis involvement is well-known in psoriatic lesions pathogenesis, and recently was described in AD, particularly in intrinsic phenotype and in Asian patients [4,29]: the use of target drugs directed against these two axes in these cases could probably lead to clinical improvement.

However, it should be noted that psoriasiform reactions have been observed in some patients treated with AD target drugs [30,31,32,33]. It is reported that in these cases, there is an increase in the expression of the Th12/23 cytokines in lesional skin, probably induced by Dupilumab [34,35].

## 3. Unmet Needs

Atopic Dermatitis is a chronic inflammatory itchy skin disease with a complex pathogenesis and a wide clinical variability among patients.

Despite recent advances in understanding of pathogenic mechanisms and in development of new drugs for AD treatment, there are still important unmet needs in the management of AD patients.

Recently a panel of experts has identified, through a Delphi consensus, three main macro-areas which require advancements: diagnosis, management and prognosis, treatment [36].

### 3.1. Diagnosis

Currently, in absence of serum markers and universally applicable criteria, the diagnosis of AD is based on the clinical manifestations. The European Task Force for Atopic Eczema ruled in 2020 that dermatologist’s experience has a diagnostic power superior to any diagnostic criteria currently available, although these latter are necessary for epidemiological research and inclusion of patients in clinical trials [37].

The Hanifin and Rajka criteria were the first to be validated in 1980 [38]. Diagnosis requires the presence of at least 3 major and 3 minor criteria. Major criteria are Pruritus, Typical distribution and morphology of lesions, presence of chronic-relapsing dermatitis and personal or family history of eczema. Validation studies report a sensitivity and specificity of 87.9–96% and 77.6–93.8%, respectively [39]. According to a meta-analysis published in 2017, they are the most used criteria in clinical trials and epidemiological studies [40].

UK Working Party criteria represent a necessary derivation of the Hanifin and Rajka criteria, which have proved some limitations in applicability outside hospital environment. They are of well-known validity and diagnosis is formulated by satisfying 1 mandatory major criterion which is represented by the appearance of pruritic dermatosis in the last 12 months or history reported, plus at least 3 additional criteria. Additional criteria includes: history of folds dermatosis, personal history of atopic manifestations, history of cutaneous xerosis in the last 12 months, dermatitis of flexural areas started before 2 years of age [41,42,43]. Validation studies report a sensitivity and specificity higher than 85% and 96%, respectively.

The American Academy of Dermatology published in 2014 guidelines for AD diagnosis and assessment, where the Hanifin and Rajka criteria have been simplified. Clinical features were classified into essential, important and associated; clinical conditions mimicking AD such as scabies, seborrheic dermatitis, psoriasis or cutaneous t cell lymphoma should be also excluded [44].

Adult onset-AD is heterogeneous: diagnosis is often a challenge.

The so far validated diagnostic criteria are not adequate for adult onset-AD because they have been developed for classic pre-adult-onset AD; specific validated diagnostic criteria for this subgroup of patients is needed to simplify the diagnosis and to include patients in clinical and epidemiological studies. Recently a panel of experts has drawn up a series of criteria for diagnosis of adult onset-AD, which include morphological characteristics, localization of lesions, clinical history and exclusion of differential diagnoses [45].

### 3.2. Management and Prognosis

Currently, serum-specific markers for AD diagnosis are not available. Furthermore, ensure that markers for prognosis and disease course assessment are not available.

Dosing total and specific IgE serum level should be often helpful in clinical practice. It is estimated that specific IgE sensitization against food or aeroallergen affects about 85% AD patient [27]: in extrinsic AD the lost of epidermal barrier integrity observed, facilitates antigens penetration in the skin and subsequently activates Th2-mediated inflammation.

However, not all patients have high levels of IgE: this condition is known as intrinsic AD. Intrinsic Atopic Eczema is characterized by epidermal barrier preservation with aptens and/or metals penetration through skin. In addition to Th2, both Th1 and Th17 axes are involved [29].

It is estimated that about 80% of AD patients can be classified into the extrinsic phenotype with elevated IgE levels, while 20% into the intrinsic phenotype with normal IgE levels [46].

Although elevated in most patients, total IgE levels are lacking in specificity. In addition, IgE levels tend to vary with disease severity, so they are not a reliable indicator since there are patients with severe disease with normal IgE values. Furthermore, they themselves may be elevated in other conditions such as parasitic infections, autoimmune diseases or cancers. Elevated levels of specific IgE have also been found in 55% of the general U.S. population [47].

There is a great deal of evidence supporting the important role of eosinophils in the AD pathogenesis, especially in the genesis of pruritus [48,49,50,51,52,53]. However, an elevated eosinophil count is not predictive of AD in the absence of skin manifestations; moreover, they are nonspecific because they could be elevated in other conditions such as Bullous Pemphigoid.

Important efforts are being made in identifying serological markers of disease to achieve personalized medicine through the use of target drugs tailored to patient characteristics.

Recently, some papers have been published with this purpose.

A study of 147 serum markers in 193 AD patients identified four clusters of patients, each one with specific serological markers. SASSAD score was used to assess disease severity. Cluster 1 is characterized by higher disease severity and BSA, high levels of pulmonary and activation-regulated chemokines (PARC), tissue inhibitors of metalloproteinases 1 (MP-1), and soluble CD14. Cluster 2 is characterized by low disease severity with the lowest levels of IFNα, MP-1 and VEGF. Cluster 3 has a high severity score with the lowest levels of IFNβ IL-1 and epithelial cytokines. Cluster 4 has a low severity score but with the highest levels of IL-1, IL-4, IL-13, and TSLP.

About 48% of patients can be classified into clusters 1 and 4, characterized by highest levels of Th2 cytokines and erythematous skin phenotype. For this reason, they should be considered the best candidates for Th2 target therapy with dupilumab, lebrikizumab, and tralokinumab. Conversely, the remainder of patients are classifiable into clusters 2 and 3, with low levels of Th2 cytokines. These groups are not considered ideal candidates for Th2 target therapy, so further molecular pathogenic characterization studies would be needed in order to develop specific target drugs [54].

In a prior work conducted on 143 markers from 146 patients with severe AD assessed by EASI score, four different clusters were identified. Cluster A is characterized by “skin homing chemokines/IL-1R1-dominant”; cluster B by “Th1/Th22/PARC-dominant”; cluster C by “Th2/TH22/PARC-dominant”; cluster D “Th2/eosinophil-inferior” with the lowest levels of eotaxin, RANTES, PARC, MDC, TARC and IFNα. In addition, the authors have identified a small panel of serologic markers that allows them to classify patients into one of the 4 original clusters using a prediction model. These markers are represented by IL-37, IL-1ra, XCL-1, eotaxin, IL-1β, IL-26, TNFSF14, IL1R1, EGF, and TSLP. Curiously, none of these latter few are Th2-related cytokines, so the authors hypothesize that, although AD is a Th2-related disease, these markers could play a key role in disease pathogenesis, and thus in treatment response.

Although it is speculated that in the future, endotype division will allow patients to be treated by target therapy tailored to their own characteristics, RCT or longitudinal clinical trials are first needed to evaluate the utility of these markers in target therapy [55]. According to a 2015 meta-analysis TARC would be a reliable marker in correlating with disease severity [56].

Another key concept is the definition of disease severity and the use of clinimetric indices that allow the clinician to quantify the disease burden. The most recent American guidelines report the SCORAD index, EASI, IGA and SASSAD as the most widely used scoring systems [44].

Furthermore, the latest Italian guidelines report SCORAD, IGA and EASI as the most used scores; moreover, EASI score should be preferred in clinical trials and daily clinical practice to assess disease severity. Specifically, moderate–severe AD is defined by EASI score ≥ 16 or <16 with one of the following features: localization in sensitive sites (face, hands, genitalia), itching NRS ≥ 7, sleep disturbance NRS ≥ 7; DLQI ≥ 10. To assess the change in AD signs, the authors suggest combined use of EASI and NRS pruritus scale; in addition, they suggest DLQI for disease impact assessment on patient [57].

### 3.3. Treatment

Despite the wide range of clinical features and knowledge about disease molecular mechanisms, treatment of Atopic Dermatitis is currently challenging. To be successful, treatment should consider several variables closely related to patient such as age, needs, extent and site of lesions, onset and disease course, possible response to previous therapies, caregiver’s education and support.

According to European guidelines published in 2018, treatment of Atopic Dermatitis involves a four-step approach, based on the EASI score. The basis of AD treatment is the identification and avoidance of any trigger factors, as well as the use of topical emollients.

Especially in moderate-to-severe AD, the use of the most common conventional immunosuppressive drugs and systemic corticosteroids is limited by short- and long-term side effects and therefore cannot be recommended for a long period.

EMA has approved Dupilumab for adults and adolescents >12 years old and children between 6–11 years old with moderate–severe AD who are candidates for systemic therapy. Dupilumab is a recombinant human monoclonal antibody which inhibits IL-4 and both IL-4 and IL-13 signal transduction targeting type I (IL-4Rα/γc) and type II receptor (IL-4Rα/IL-13Rα), respectively. This results in reduced levels of Th2 biomarkers TARC and total and specific IgE. Dupilumab demonstrated high efficacy and safety in phase II and III studies; nevertheless, only about 40% of patients achieved complete or near-complete response, with or without concomitant corticosteroid use [58,59,60].

Dupilumab has changed the paradigm of treatment of severe atopic dermatitis.

A multicenter case series of 19 adolescents affected by moderate–severe AD who started dupilumab during COVID 19 Pandemic Outbreak in early 2020 was conducted. All patients reached EASI-50 and EASI-75 was reached in 78.9% of cases; EASI-100 was achieved in 15.8% of patients. Reduction of 77.5% of cDLQI and decreasing of 5.9 points of NRS itch score have been reached. Efficacy and safety were excellent, with one patient who contracted asymptomatic SARS-CoV2 infection and 1 who developed mild conjunctivitis (5.3%) [61].

In a retrospective observational study, patients achieve EASI 75 and EASI 90 at w16 in 66.7% and 22.2%, respectively, confirming that dupilumab is an effective and well-tolerated treatment for moderate–severe AD in adolescents [62].

A real-world experience assessed the efficacy and safety of Dupilumab in pediatric AD patients. Fifteen pediatric moderate–severe AD patient were enrolled. EASI score was 19.23 ± 3.03 at baseline and 1.69 ± 0.54 at following up for 6 months after standardized treatment protocol. CDLQI was 13.53 ± 2.88 at baseline and 1.60 ± 0.63 at following up for 6 months after standardized treatment protocol. The most frequent adverse event was conjunctivitis. No serious adverse events occurred during the treatment period [63].

Data from a French multicenter retrospective cohort showed a significant decrease a significant decrease in SCORAD (mean: 21.8 ± 13.8 vs. 53.9 ± 18.5; *p* < 0.0001) and IGA (1.3 ± 0.8 vs. 3.5 ± 0.7; *p* < 0.0001) after 3 months of treatment. Adverse events have been reported: conjunctivitis (11.3%); dupilumab facial redness (3 patients); injection site reactions (17.5%). Treatment interruption was observed in 6.3% [64].

The Italian group conducted a multicenter study to evaluate the effectiveness and safety of dupilumab in the treatment of children aged from 6 to 11 years in real life experience. A total of 55 AD children (24 males [43.64%], 31 females [56.36%]; mean age 9.35 ± 1.75 years) were included. EASI score, P-NRS, S-NRS and c-DLQI have been used to assess disease severity. At W16 the proportion of patients achieving EASI75 was 74.54% and also a significant mean percentage reduction for P-NRS, S-NRS and c-DLQI was reached (68.39%, 70.22% and 79.03%, respectively) [65].

Tralokinumab is a target human monoclonal antibody recently approved by EMA for moderate-to-severe AD treatment in adult patients who are candidates for systemic therapy. The drug binds and inhibits IL-13 and prevents its interaction with its own IL-13α1/IL-4Rα receptor. The result is lowered levels of Th2 serum biomarkers in the blood and lesional skin of patients. In the randomized, double-blind, placebo-controlled, phase III monotherapy trials, EASI75 was achieved in 41.6–49.9% of patients at 16 weeks and in 49.1–51.4% of responders at 52 weeks, respectively [66].

In a double-blind, randomized, multicenter, placebo-controlled phase III study with concomitant use of corticosteroids, EASI75 was achieved in 56% of patients; of these, about 90% maintained EASI75 at 32 weeks [67].

Itch reduction and lesion remission are neither uniform nor total among different patients: the big unmet need is the fact that there is a substantial percentage of patients who do not respond optimally to the therapies approved so far, for whom more effective alternative target treatment are needed. In addition, subcutaneous administration may not be preferred by the patient or otherwise the most appropriate. Therefore, development of additional target therapeutic strategies is needed.

Great research interest has been generated by JAK/STAT signaling pathway inhibitors.

Janus kinases are enzymes that transduce intracellular signals from receptors on the cell surface involved in various functions including immune response and inflammation. JAKs phosphorylate and activate signal transducers and activators of transcription (STATs) that downstream induce expression of specific genes within the cell. With the partial block of this signal way, JAK inhibitors modulate the downstream signaling pathway.

Oral Baricitinib is a JAK 1–2 inhibitor which has been approved by EMA for adult patients with moderate-to-severe AD who are candidates for systemic therapy [68]. In randomized, double-blind, placebo-controlled phase III clinical trials, Baricitinib proved efficacy and safety with an EASI75 response rate of 17.9–24.8% in monotherapy and 43.1–47.7% in combination with corticosteroids at 16 weeks, respectively [69,70]. The most commonly reported side effects were nasopharyngitis, upper respiratory tract infections and folliculitis.

Abrocitinib is an oral selective JAK1 inhibitor that proved to be effective and safe in monotherapy in the clinical trial conducted in patients older than 12 years with moderate-to-severe AD [71]. EASI75 was achieved in 61% and 44.5% of patients treated with 200 mg and 100 mg Abrocitinib at week 12, respectively. Adverse events have been reported in about 60% of treated patients. An additional phase III clinical trial demonstrated the superiority of oral Abrocitinib in itch improvement compared with the dupilumab-treated patient group at week 2 [72].

Oral Upadacitinib, a selective JAK2 inhibitor has recently received an indication for treatment of severe AD in individual older than 12 years who are candidates for systemic therapy. In a phase IIb multicenter, randomized, placebo-controlled, double-blind dose-ranging clinical trial conducted in patients with moderate–severe AD, it demonstrated efficacy in reducing pruritus [73].

In a phase III study on adult and adolescent >12 years old with moderate–severe AD, EASI75 response rate of 38.1% and 50.6% was achieved in patients treated with Upadacitinib 15 mg and 30 mg, respectively. Acne and Upper respiratory tract affections were the most common side effects reported [74,75].

Although JAK inhibitors are very expensive, the economic loss due to severe moderate AD is also important. A cross-sectional survey published in 2020 conducted in the 5 major European countries (Italy, Germany, France, Spain, UK) showed a total annual direct cost per patient per year of EUR 2242–EUR 6924 and an annual indirect cost of EUR 7277–EUR 14,236, dependent on the level of disease severity. In particular, sleep disorders, anxiety and depression, were reported in 61.6%, 52.7% and 75.5% of patients, respectively, increasing work impairment [76]. Despite high costs, oral JAK inhibitors have been shown to rapidly improve itching and skin rash in patients with moderate–severe AD with minimal side effects: this may represent a game changer for the treatment of these challenging patients [77].

Adult atopic dermatitis (adult AD) is a systemic inflammatory disorder, whose relationship with immune-allergic and metabolic comorbidities is not well established yet.

The relationship between adult AD and metabolic comorbidities remains unclear.

According to the “inflammatory skin march model” there is a link between metabolic abnormalities and AD in adults via cytokines of Th1, Th17 and Th22 families [78].

The results from a nationwide multicenter Italian study [79] show that atopic comorbidities (Asthma, Conjunctivitis, Rhinitis) were significantly associated with having moderate-to-severe AD in young adult patients.

According to their data, oral corticosteroids and cyclosporine were the most widely used immunosuppressant despite the unfavorable risk/benefit ratio of systemic corticosteroids long-term use for adult patients.

Systemic comorbidities such as hypertension or diabetes could influence therapeutic choice in adult patients who are affected by moderate–severe AD.

Treatment atopic dermatitis requires standardization among clinicians: further studies are needed to evaluate the complex relationship between AD severity, comorbidities and therapeutic choices.

Currently, a clear definition of remission is not available for AD: therapeutics goal needs to be well defined. Additionally, the differentiation between short-, medium- and long-term goal is still a unmet need in AD management [80].

Four outcome domains to be measured in AD clinical trial has been established by the Harmonizing Outcome Measures for Eczema (HOME) initiative: clinician-reported signs, patient-reported symptoms, quality of life and long-term control [81,82,83].

In previous consensus the experts individuate EASI and POEM as the preferred instruments for clinician-reported signs; in addition, in 2021 they agreed that QoL should be measured using the DLQI, CDLQI and IDQoL for adults, children and infants, respectively [84].

Achieving disease control or at least a clinically significant improvement in its activity is considered the goal of systemic therapy; nevertheless, maintaining remission of both symptoms and eczematous lesions could be challenging. According to the European Task Force on Atopic Dermatitis a treat-to target approach is recommended: multidisciplinary assessment is important to identify patient relevant treatment targets [85].

Developing a shared treatment strategy is important for AD: obtaining a long-term remission is a shared target in AD management.

Recently a group of experts provided a set of recommendations and algorithm for decision-making in treating moderate-to-severe atopic dermatitis (AD) to target with systemic drugs [86].

More recently, a consensus of SIDeMAST, AAIITO, ADOI, SIDAPA and SIAAIC has proposed an additional treatment algorithm for AD. They assumed DLQI and or ADCT as patient impact measure and EASI and/or NRS pruritus as Disease measure. According to the statement, continuation of therapy should be considered if treatment goals are met for at least one measure of impact plus at least one measure of disease, while therapy should be modified if neither patient impact measure nor disease measure goal is achieved. In addition, treatment should be discussed and optimized in agreement with patient’s preference if whether an patient impact measure or a disease measure is met [59].

## 4. Conclusions

Despite recent advances in understanding pathogenic mechanisms and identifying patient phenotypes and endotypes, there are still numerous unmet needs in atopic dermatitis. Actually, AD diagnosis is clinical. Validated diagnostic criteria that are applicable in clinical practice are needed to increase diagnostic accuracy as serologic markers to guide diagnosis and establish disease prognosis. Considering the variability in patients’ manifestations and needs, the use of clinimetric indices is useful in clinical practice to assess the course and activity of disease; in particular, the use of EASI, pruritus NRS, and DLQI is suggested. Currently, we are facing benevolent times for patients with atopic dermatitis: in fact, several drugs have been approved for use in atopic dermatitis and more are under study. The use of oral JAK inhibitors may be a game changer for patients with moderate–severe AD. However, it is important to define treatment goals in a way that optimizes efforts to achieve harmonization of treatment strategies. A multidimensional approach is therefore needed.

## Data Availability

No new data were created or analyzed in this study.

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
