# Peer review of "Atopic Dermatitis: Clinical Aspects and Unmet Needs"

_biomedicines, 2022, doi:10.3390/biomedicines10112927_

Round 1

Reviewer 1 Report

This review article entitled “ ATOPIC DERMATITIS: CLINICAL ASPECTS AND UNMET NEEDS” is informative from the clinical point of view, but there are some concerns that need to be addressed as follows.

Major Concerns

The introduction session is not adequately prepared to provide current information on atopic dermatitis for readers.  

The authors should introduce the cellular and molecular players of atopic dermatitis focusing on neuroimmune interactions. These include activation of sensory neurons that drive itch sensations and scratching behaviors by infiltration and recruitment of neutrophils, TSLP-elicited basophils/CD4+T cells (chronic itch) and IgE-R+ basophils (itch flares) after skin barrier disruption.

Author Response

We would like to thank the reviewer for the useful suggestion, we have added the most recent findings related to the pathogenesis of acute and chronic pruritus in patients with AD, particularly the role of basophils.  (lines 15-36 of introduction section)

Reviewer 2 Report

Intriguing and well written manuscript adressing the unmet needs in atopic dermatitis.

Some suggestions:

1. Dupilumab has changed the paradigm of treatment of severe atopic dermatitis, thus the authors should provide real life data on efficacy and safety 

2. Among unmet medicla needs, authors should report how systemic comorbidities can influence therapeutic choices for patients affected by severe atopic dermatitis (Comorbidities and treatment patterns in adult patients with atopic dermatitis: results from a nationwide multicenter study.

Campanati A, Bianchelli T, Gesuita R, Foti C, Malara G, Micali G, Amerio P, Rongioletti F, Corazza M, Patrizi A, Peris K, Pimpinelli N, Parodi A, Fargnoli MC, Cannavo SP, Pigatto P, Pellacani G, Ferrucci SM, Argenziano G, Cusano F, Fabbrocini G, Stingeni L, Potenza MC, Romanelli M, Bianchi L, Offidani A; and collaborators.) 

Author Response

Dear Reviewer that you very much for the interesting suggestion . We have added data on the real life experience of dupilumab (lines 20-47 of treatment section) and we have reported how comorbidities could influence disease severity and therapeutic choice in patients with moderate-severe AD(lines 103-118 of treatment section) as asked .

Reviewer 3 Report

The study is of interest and underlines the way a clinician should think. The information studied and reported are useful to a specialist doctor to guide him in diagnosis and treatment. However, the article does not bring anything new in this field.

Author Response

Dear reviewer  The purpose of this paper was to  review the principal aspects of the diagnosis and therapy of a multifaceted disease as AD . The Ain was not to  add new evidences, but to collect existing notions about atopic dermatitis  and as you pointed  to be useful  to a specialist doctor to guide him in diagnosis and treatment. 

Round 2

Reviewer 1 Report

The paper is now acceptable for publication.